# A reconnaissance survey of farmers' awareness of hypomagnesaemic tetany in UK cattle and sheep farms

Diriba B. Kumssa[1]☺*, Beth Penrose[2]☺, Peter A. Bone[3], J. Alan Lovatt[4], Martin R. Broadley[1], Nigel R. Kendall[5‡], E. Louise Ander[6‡]

1 School of Biosciences, University of Nottingham, Sutton Bonington Campus, Leicestershire, United Kingdom, 2 School of Land & Food, University of Tasmania, Tasmania, Australia, 3 Livestock and Grassland Mineral Consultancy, Fairford, Gloucestershire, United Kingdom, 4 Institute of Biological, Environmental and Rural Sciences (IBERS), Aberystwyth University, Aberystwyth, United Kingdom, 5 School of Veterinary Medicine and Science, University of Nottingham, Leicestershire, United Kingdom, 6 Inorganic Geochemistry, Centre for Environmental Geochemistry, British Geological Survey, Keyworth, Nottinghamshire, United Kingdom

☺ These authors contributed equally to this work.
‡ These authors also contributed equally to this work.
* Diriba.Kumssa1@nottingham.ac.uk

**Data Availability Statement:** All relevant data are within the paper and its Supporting Information files.

## Abstract

Hypomagnesaemic tetany (HypoMgT) in ruminants is a physiological disorder caused by inadequate intake or impaired absorption of magnesium (Mg) in the gut. If it is not detected and treated in time, HypoMgT can cause the death of the affected animal. A semi-structured questionnaire survey was conducted from July 2016–2017 to assess farmers' awareness of HypoMgT in cattle and sheep in the UK. The questionnaire was distributed to farmers at farm business events and agricultural shows, and through a collaborative group of independent veterinary practices to their clients. Farmers were asked about (i) the incidence of presumed HypoMgT (PHT); (ii) their strategies to treat or prevent HypoMgT; (iii) mineral tests on animals, forage and soil, and (iv) farm enterprise type. A total of 285 responses were received from 82 cattle, 157 mixed cattle and sheep, and 46 sheep farmers, of whom 39% reported HypoMgT in their livestock, affecting 1–30 animals. Treatment and/or prevention against HypoMgT was reported by 96% respondents with PHT and 79% of those without. Mineral tests on animal, forage, and soil was conducted by 24%, 53%, and 66% of the respondents, respectively, regardless of PHT. There was a highly significant association between the use of interventions to tackle HypoMgT and the incidence of PHT ($p < 0.01$). The top three treatment/prevention strategies used were reported as being free access supplementation (149), in feed supplementation (59) and direct to animal treatments (drenches, boluses and injections) (45) although these did vary by farm type. Although some (9) reported using Mg-lime, no other pasture management interventions were reported (e.g., Mg-fertilisation or sward composition). Generally, the results indicate that UK farmers are aware of the risks of HypoMgT. A more integrated soil-forage-animal assessment may improve the effectiveness of tackling HypoMgT and help highlight the root causes of the problem.

**Funding:** This work was supported by the Biotechnology and Biological Sciences Research Council (grant numbers BB/N004280/1 [ELA], BB/N004302/1 [MRB], BB/N004272/1 [AL]) and the Natural Environment Research Council, through the jointly funded initiative Sustainable Agriculture Research and Innovation Club (SARIC). The project is entitled Magnesium Network (MAG-NET): Integrating Soil-Crop-Animal Pathways to Improve Ruminant Health, https://gtr.ukri.org/projects?ref=BB%2FN004302%2F1.

**Competing interests:** The authors have declared that no competing interests exist.

## Introduction

Hypomagnesaemic tetany (HypoMgT), also known as grass tetany or grass staggers is caused by either an inadequate magnesium (Mg) supply to ruminants or reduced absorption of Mg in the rumen [1]. Signs of HypoMgT can include excitability, grinding of the teeth, salivation, a lack of coordination of muscle movement (e.g., a staggering gait), lying down or muscle spasms, and can lead to death within hours [2, 3]. Hypomagnesaemia can create a safety issue for farm workers, a reduction in milk fat content [4], a higher susceptibility to hypocalcaemia [5] and dark cutting in sheep [6] and cattle carcass [7].

Historically, HypoMgT has been an issue in temperate regions around the world. A study in the 1960s of 477 New Zealand dairy farms found rates of HypoMgT to be between 0.2 and 3.9% [8] and deaths alone from HypoMgT were estimated to cost US$70 million per year in the 1970s [9]. A study in the 1990s in Northern Ireland found deficient blood serum magnesium levels ($<0.6$ mmol $L^{-1}$) in 2% of dairy cows tested (n = 3626 cows, 377 herds) and 7.3% of beef suckler cows (n = 6664 cows, 772 herds; [10]. In a UK study, 19% of dairy farms (n = 23) and 23% of suckler beef farms (n = 89) had presumed HypoMgT (PHT) in 1995 [11].

More recent data are difficult to come by, however, estimates for the temperate regions of Australia are that 60% of cattle holdings are affected by HypoMgT, and the costs of prevention, treatment of HypoMgT and the effect on cattle production are estimated to be around AUD $23 million [12]. In UK grazing ruminants, hypomagnesaemia-related conditions have been reported to affect 1–4% of cows, rising to 20–30% within individual herds [13]. To the authors' knowledge, there are no recent estimates of the prevalence of HypoMgT in UK livestock systems.

Preventative measures to reduce the risk of HypoMgT include: (i) direct to animal strategies such as administering Mg-containing bullets or boluses, and delivering Mg via drenches; (ii) in-feed strategies where Mg is included in a concentrate or total mixed ration, usually as part of a mineral premix; (iii) adding Mg to the drinking water; (iv) providing free-access Mg-containing salt licks and buckets; and (v) soil or plant-based strategies. Soil and plant-based strategies in particular are wide-ranging, and can include foliar and soil-based application of Mg fertilisers, dusting pasture with Mg-containing minerals [1, 14], application of K fertilisers during winter, applying nitrogenous fertiliser with Mg if nitrogen is applied to pasture before grazing, grazing on permanent pastures in spring, and cutting and removing grass before grazing on paddocks with a grass tetany history [15]. Direct treatment of HypoMgT is usually via subcutaneous injection of magnesium sulphate [14] or magnesium hypophosphate often alongside calcium borogluconate.

A 1991 survey in Northern Ireland found that 19.8% of dairy (n = 263 farmers) and 20.9% of (n = 522 farmers) beef suckler farmers surveyed reported that they did not use any form of Mg supplementation [10]. Roderick et al. [11] reported a 19% and 23% incidence of grass staggers in dairy and beef suckler herds, respectively under organic farming systems in the UK. To the authors' knowledge, there are no up-to-date data available regarding the extent to which preventative or treatment measures are currently being used in the UK.

The purpose of this study was to: (i) provide a snapshot of the extent of Mg deficiencies in UK dairy, beef and sheep enterprises; (ii) record the measures UK livestock producers were taking to treat or prevent HypoMgT in their herds or flocks; (iii) determine how aware UK livestock producers were of Mg concentrations in their soil, pasture and livestock; and (iv) understand any associations between the incidence of Mg deficiencies, awareness of their vulnerability to Mg deficiencies, and the strategies they were employing to reduce the risk of HypoMgT.

## Materials and methods

A semi-structured questionnaire (see S1 Information) survey was used to collect data on the prevalence of HypoMgT in UK cattle and sheep farms (Fig 1). Ethical approval was sought before conducting the survey (Ethical approval number 1814160621, School of Veterinary Medicine and Science Ethical review panel, University of Nottingham). Data collection was carried out either online using KoBoToolBox or by distributing paper questionnaires to farmers at industry events including AgriScot, and the Royal Welsh Summer and Winter shows, and Agriculture and Horticulture Development Board (AHDB) farm knowledge exchange events between July 2016–2017. The questionnaires were also distributed to clients of veterinary practices within XLVets, a collaborative group of independent veterinary practices in the UK. Prior to completing the survey, farmers were presented with information sheet (see S1 Information) about the objective of the survey and how data collected during this survey was to be used. Where paper questionnaires were handed to respondents at farm events or via their vet, responses were received via postal mail. Some farmers filled out digital forms at the event. Paper-based responses were input into KoBoToolbox for data compilation. The information sheet and survey were made available in both English and Welsh in both paper and digital form, and participants could respond in either language. The Welsh language responses were translated into English before interpretation of the data. Data were collected on (i) the incidence of presumed HypoMgT (PHT); (ii) ways used to treat HypoMgT; (iii) mineral testing conducted on animals, forage or soil; and (iv) farm type and herd or flock size.

Data cleaning was carried out in Microsoft Excel 2016. Thirteen questionnaire responses were excluded where the number of cattle was <11, the number of sheep was <26, and where there were contradictory responses. Statistical analysis was conducted using IBM SPSS 25. Pearson's chi square test was used to check the association between incidence of PHT, mineral testing and use of treatments against HypoMgT. Farms were grouped into beef, dairy and sheep farm enterprise and combinations to assess if there was variation among enterprise types with regards to incidence of HypoMgT, mineral tests and treatments to control HypoMgT, where there were 30 or more respondents from a farm enterprise group. Euler diagrams were drawn using an online tool at http://bioinformatics.psb.ugent.be/webtools/Venn/. Map was produced using ArcMap 10.7. Other visualisations were produced using Tableau Desktop (Professional Edition 2019.2).

## Results

A total of 285 useable responses were received from Wales (51%), England (35%), Scotland (11%) and Northern Ireland (3%). When considered by ruminant enterprise (i.e., beef, dairy and sheep), 45% of the respondents had mixed beef and sheep enterprises, 13% had dairy only and 16% had sheep only enterprises. The remaining ~26% of the respondents were beef only, mixed beef-dairy, mixed beef-dairy-sheep, and mixed dairy-sheep enterprises, with the number of respondents in each category and combination less than 30 farms (Fig 2). Compared to the nationally representative Farm Business Survey [16], this survey's respondents were skewed towards beef and sheep farmers, although this also reflects the geographic location of participants in the west of Britain where most beef and sheep production is concentrated [17].

### Prevalence of hypomagnesaemic tetany (HypoMgT)

Out of 285 respondents, 110 farmers (39%) stated their livestock had been affected by HypoMgT at some point (Fig 3b). By farm enterprise, 47% of mixed beef and sheep, 33% of sheep and 32% of dairy farm participants reported incidence of HypoMgT in their livestock affecting 1 to 30 animals in a given year (Fig 3a). There was no significant association between



**Fig 1. Number of respondents by district.** Geographical distribution of cattle and sheep farmers that participated in the survey. Number of respondents presented by district to protect individual farm identity.

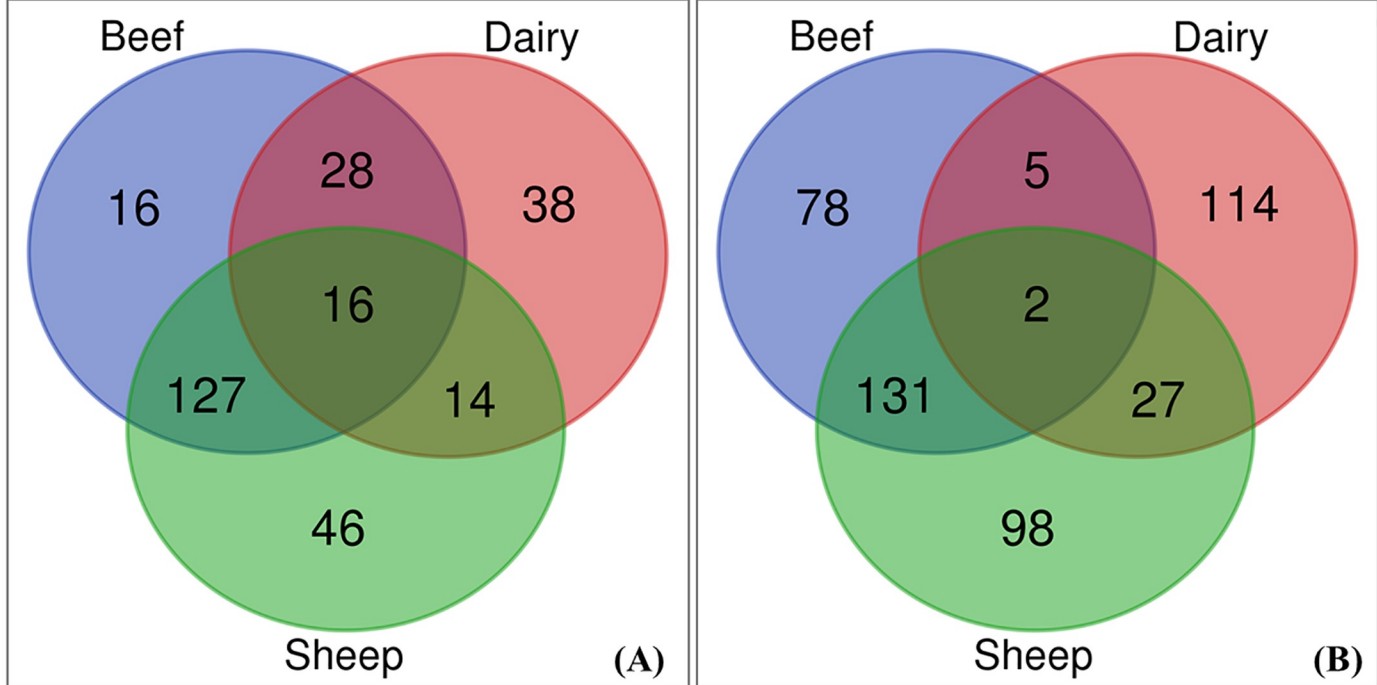

**Fig 2. Survey respondents by farm enterprise and combinations. (A)** Number of beef, dairy, and sheep farms and their combinations in the current survey, n = 285. **(B)** Number of beef, dairy, and sheep farms in the 2012/13 farm business survey, n = 455.

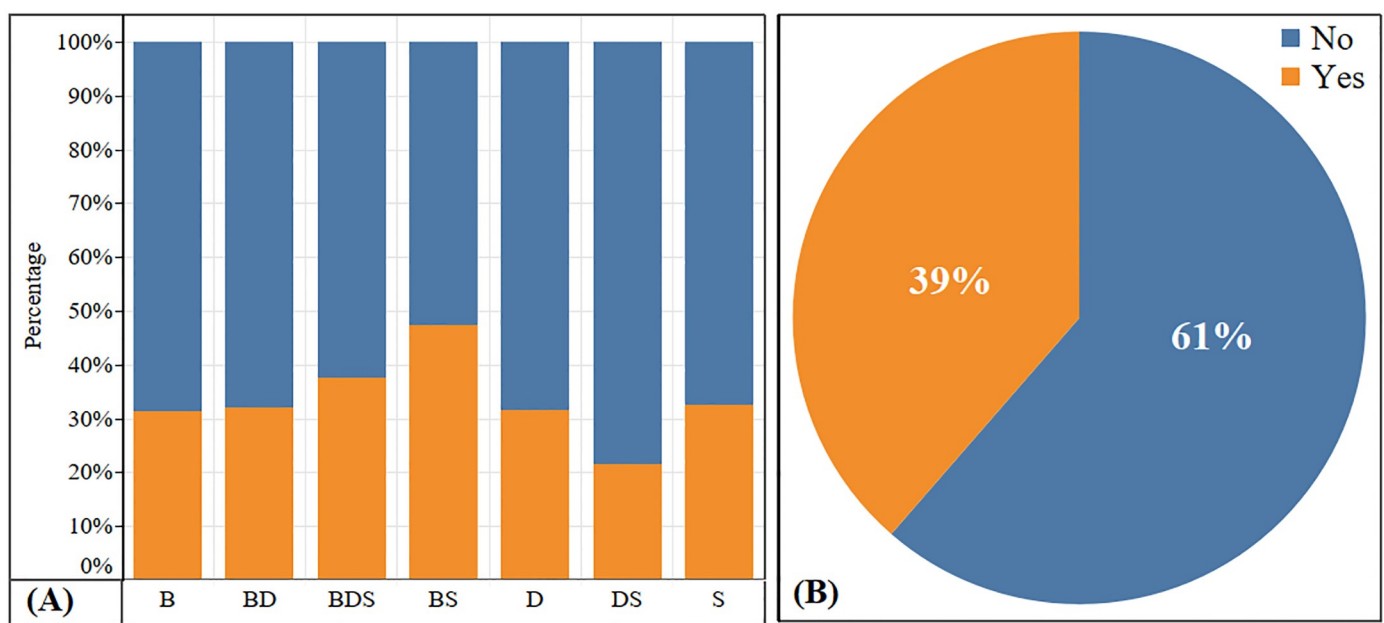

**Fig 3. Incidence of presumed hypomagnesaemic tetany (PHT). (A)** Percentage of beef (B), mixed beef-dairy (BD), mixed beef-dairy-sheep (BDS), mixed beef-sheep (BS), dairy (D), mixed dairy-sheep (DS), and sheep (S) farms that reported incidence (yes) or no incidence (no) PHT. **(B)** Overall percentage incidence of PHT.

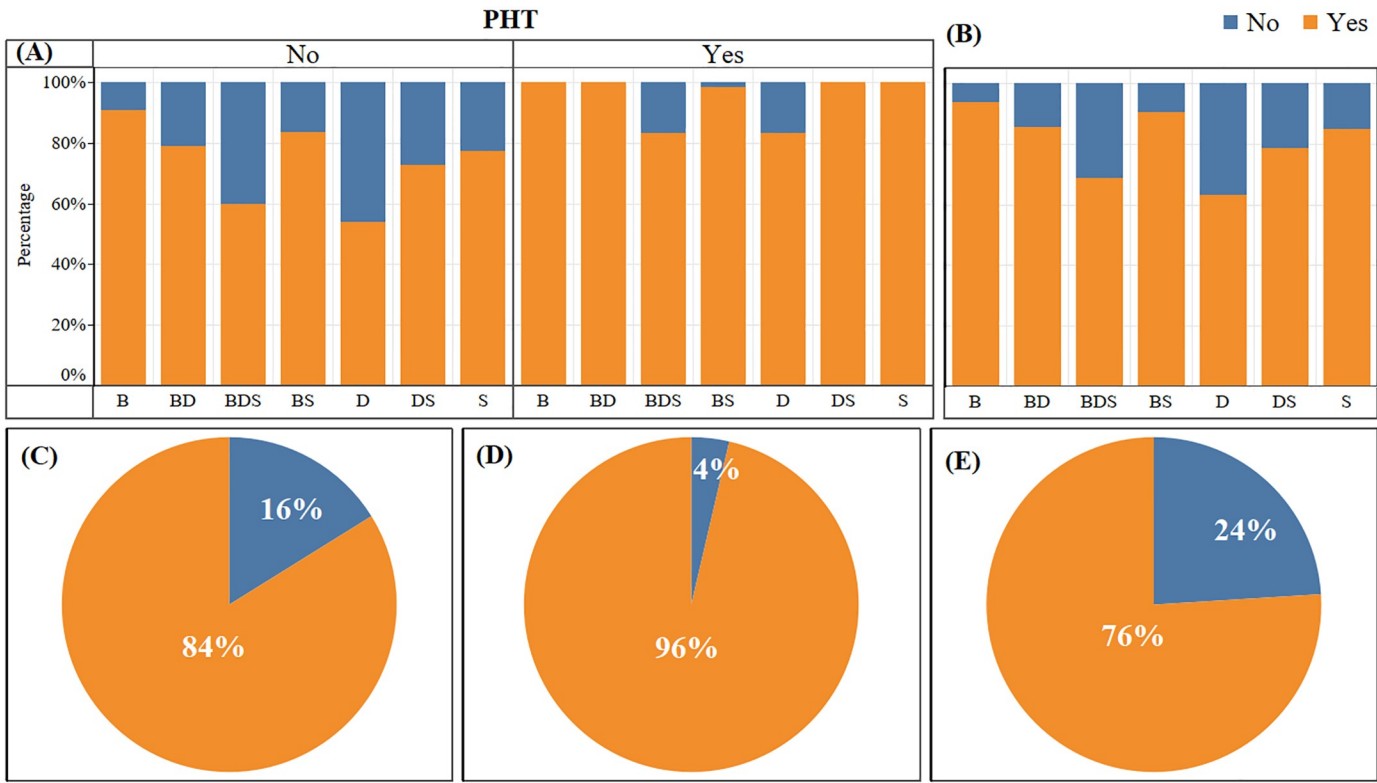

**Fig 4. Use of interventions to tackle hypomagnesaemic tetany (HypoMgT).** Percentage of beef (B), mixed beef-dairy (BD), mixed beef-dairy-sheep (BDS), mixed beef-sheep (BS), dairy (D), mixed dairy-sheep (DS), and sheep (S) farms that used (Yes) or did not use (No) Mg treatment measures to tackle hypomagnesaemic tetany (HypoMgT) **(A)** with (Yes) and without (No) the incidence presumed HypoMgT (PHT) and **(B)** regardless of the incidence of PHT. **(C)** Overall percent usage of interventions regardless of presumed hypomagnesaemic tetany (PHT) incidence, **(D)** with PHT incidence and **(E)** without PHT incidence.

the prevalence of PHT and farm enterprise types (n = 285, $\chi^2$ = 8.1, degrees of freedom [d.f.] = 6, $p$ = 0.2).

## Interventions to tackle hypomagnesaemic tetany (HypoMgT)

Irrespective of the incidence of presumed HypoMgT (PHT), 84% of the respondents had used preventative and/or treatment measures against HypoMgT (Fig 4b and 4c). These interventions were categorised into in-feed, in-water, free access supplements (licks, blocks and buckets), direct (boluses, drenches and injections), pasture dressing (e.g., Mg lime/fertiliser), management changes (e.g., avoiding certain fields at certain times), a generic mineral supplementation (route of supplementation not indicated but likely to be either in feed, free access or direct) or nothing (Tables 1 and 2). A total of 308 intervention categories were reported by the 226 participants who did something and the most common intervention reported was use of free access minerals (n = 149; 62% of the number of respondent who carried out a treat/prevent strategy). The next most common interventions were in feed strategies (n = 59, 25%) and direct to animal strategies (n = 45, 19%) with all others categories having n<25. There were 10 farms which stated prevention/treatment strategies, which would not address a Mg issue: generally direct-to-animal trace element supplements that do not contain Mg. Only 3% (n = 9) of the respondents reported the use of Mg-lime to correct underlying available soil Mg-deficiencies. Out of the 110 farmers who reported incidence of HypoMgT, 96% reported using treatment/prevention interventions to abate it (Fig 4a and 4d). Similarly, out of the 175 farmers

**Table 1. The distribution of interventions used, according to species and incidence of presumed hypomagnesaemic tetany (PHT).**

| Farm | PHT | In feed | Free access | Generic | In water | Direct | Pasture | Management | Nothing |
|---|---|---|---|---|---|---|---|---|---|
| Cattle | All | 27 | 25 | 5 | 13 | 10 | 2 | 2 | 19 |
| | No | 18 | 15 | 3 | 6 | 3 | 1 | 1 | 17 |
| | Yes | 9 | 10 | 2 | 7 | 7 | 1 | 1 | 2 |
| Cattle and sheep | All | 26 | 100 | 6 | 10 | 27 | 6 | 6 | 22 |
| | No | 16 | 49 | 5 | 5 | 10 | 5 | 1 | 20 |
| | Yes | 10 | 51 | 1 | 5 | 17 | 1 | 5 | 2 |
| Sheep | All | 6 | 24 | 1 | 1 | 8 | 1 | 2 | 5 |
| | No | 4 | 14 | 1 | | 3 | | 2 | 5 |
| | Yes | 2 | 10 | | 1 | 5 | 1 | | |
| | Grand total | 59 | 149 | 12 | 24 | 45 | 9 | 10 | 46 |

**Table 2. Number of farms that used various interventions to prevent/treat hypomagnesaemic tetany (HypoMgT) by categories for the different species and incidence of presumed HypoMgT (PHT).**

| Interventions (n) | PHT | Cattle | Cattle and Sheep | Sheep |
|---|---|---|---|---|
| None | Yes | 2 | 2 | 0 |
| | No | 17 | 20 | 5 |
| One | Yes | 14 | 46 | 6 * |
| | No | 28 | 48 * | 15 |
| Two | Yes | 9 | 16 * | 4 |
| | No | 8 | 19 * | 2 * |
| Three or more | Yes | 2 | 3 | 2 |
| | No | 1 | 2 | 1 |

* Two further interventions which were inappropriate to treat/prevent HypoMgT. Total n = 285, cattle n = 82, cattle and sheep n = 157, sheep n = 46.

that reported no incidence of HypoMgT in their livestock, 76% had used similar range of interventions as preventative and/or treatment measures (Fig 4e). Overall, 42 farms without PHT and 4 farms with PHT did not use any HypoMgT intervention. There was an overall highly significant association between the use of interventions to tackle HypoMgT and the incidence of PHT (Table 3) with variation among farm enterprises. In the dairy enterprises, there was no significant association between incidence of PHT and use of intervention. In the sheep farming enterprises, the association was marginal (Table 3).

## Animal, forage and soil mineral testing

Overall, 24%, 53% and 66% of all the respondents reported mineral testing on animal, forage, and soil, respectively (Fig 5g–5i). Small proportional differences occurred where PHT was

**Table 3. Fisher's exact test of association among the use of interventions to prevent/treat hypomagnesaemic tetany (HypoMgT), incidence of presumed HypoMgT (PHT) and farm enterprises.**

| Farm enterprise | n | Exact significance (2-sided) |
|---|---|---|
| Beef-sheep | 127 | 0.005 |
| Dairy | 38 | 0.147 |
| Sheep | 46 | 0.078 |
| Total | 211 | 0.000 |

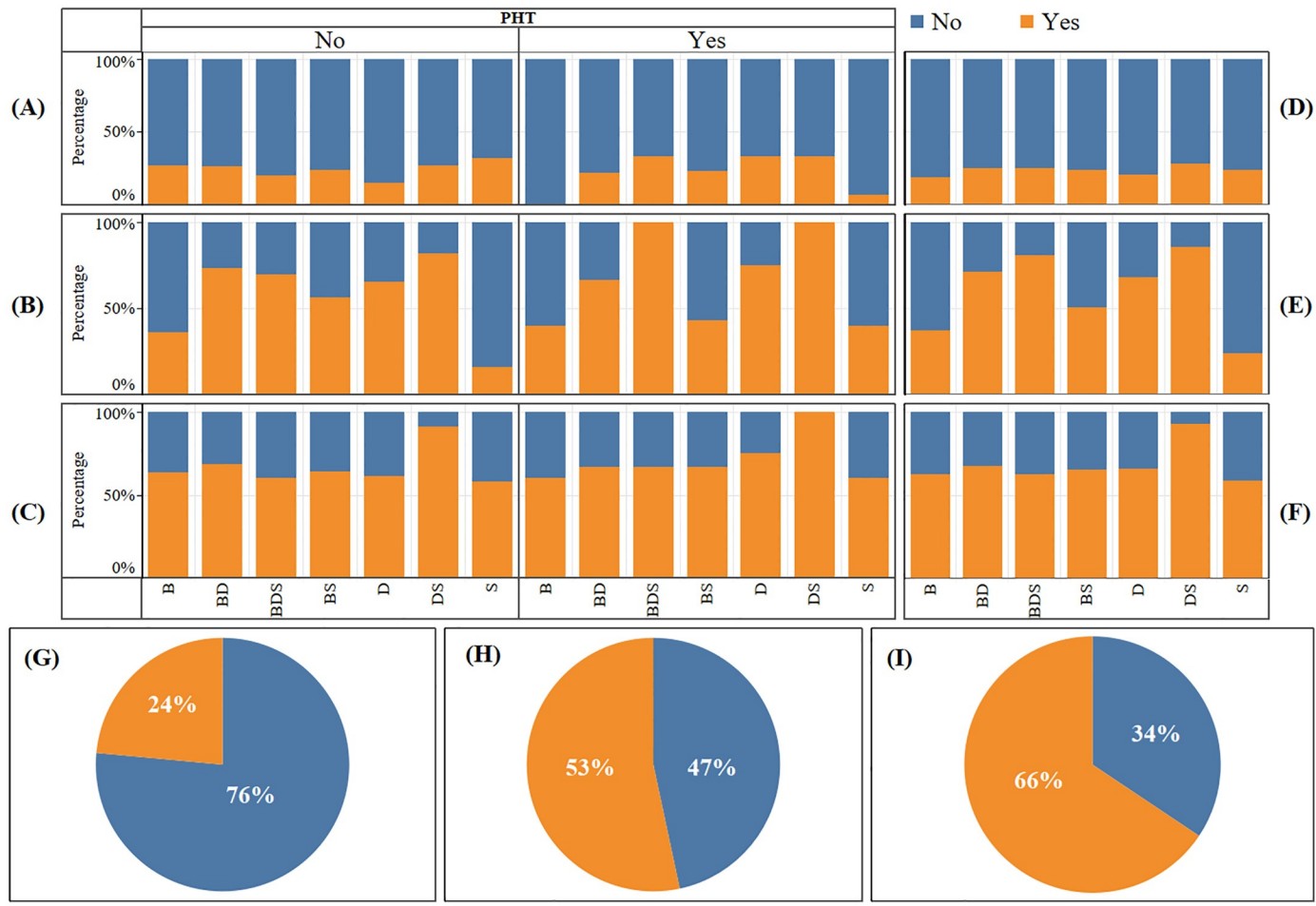

**Fig 5. Mineral testing by farm enterprise and incidence of presumed hypomagnesaemic tetany (PHT).** Percentage of beef (B), mixed beef-dairy (BD), mixed beef-dairy-sheep (BDS), mixed beef-sheep (BS), dairy (D), mixed dairy-sheep (DS), and sheep (S) farms that conducted (Yes) or did not conduct (No) mineral tests on: **(A)** animal, **(B)** forage and **(C)** soil with and without the incidence of presumed hypomagnesaemic tetany (PHT). Percentage of farm enterprises that conducted (Yes) and did not conduct (No) mineral tests on: **(D)** animal, **(E)** forage and **(F)** soil regardless of the incidence of PHT. Overall percentage of farms that conducted (Yes) and did not conduct (No) mineral tests on: **(G)** animal, **(H)** forage and **(I)** soil.

reported, 25%, 54%, and 65%, and was not reported, 22%, 53% and 67%, for animal, forage and soil mineral tests, respectively (Fig 6b). Soil mineral tests were reported by 59%, 65% and 68% of the sheep, mixed beef and sheep, and dairy farm respondents, respectively (Fig 7f) whilst 24%, 50% and 68%, of the same enterprise types used forage mineral tests (Fig 7e). Direct testing of livestock was used at 24%, 21% and 24%, of the sheep, mixed beef and sheep, and dairy farms, respectively (Fig 7d). Overall, 64 respondents reported using no mineral testing at all (Fig 6).

Mineral testing on animals, forage and soil was independent of the incidence of PHT (Table 4). Forage mineral testing was conducted proportionally more by respondents with dairy and dairy-mixed enterprises compared to the other enterprises (Fig 5b and 5e). When there was no incidence of PHT, 65–82% of the dairy and mixed-dairy farms compared to 16–57% of the beef and /or sheep farms conducted mineral testing on forages. With the incidence of PHT, 67–100% of the dairy and mixed-dairy farms compared with 40–43% of the beef and/or sheep farms carried out mineral tests on forages (Fig 5b). Among the sheep farmers, 40% of

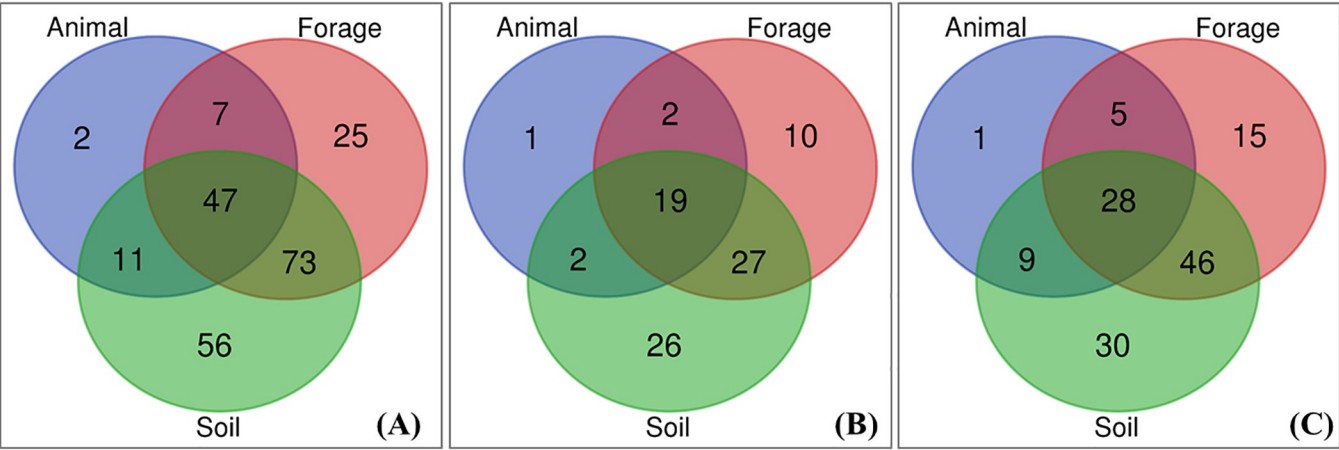

**Fig 6. Number of cattle and sheep farms that conducted mineral tests on animal, forage and soil. (A)** Overall. Those not testing = 65. **(B)** Farms reporting hypomagnesaemic tetany (HypoMgT). Those not testing = 23. **(C)** Farms reporting no HypoMgT. Those not testing = 42. Total n = 285.

the respondents with PHT conducted forage mineral tests compared to 16% of the farmers who reported no PHT (Fig 5b).

## Discussion

The aim of this study was to determine the awareness of ruminant Mg deficiency among UK farmers. Due to the nature of recruitment, the results may not be representative of the wider grazing ruminant industry as participant selection for this study was carried out following a convenience sampling approach, rather than through a representative survey. Thus, the proportions reported should not be taken as representative of the grazing sector as a whole, or of the specific farm enterprise types included. It is likely that the farmers who participated in the current survey were particularly engaged, for example, seeking scientific and business information at agricultural events and/or shows to improve their livestock health, productivity and management.

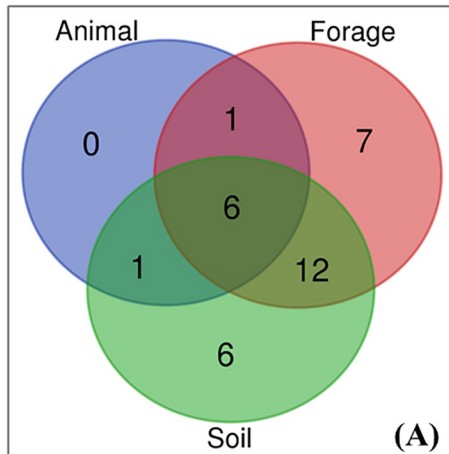 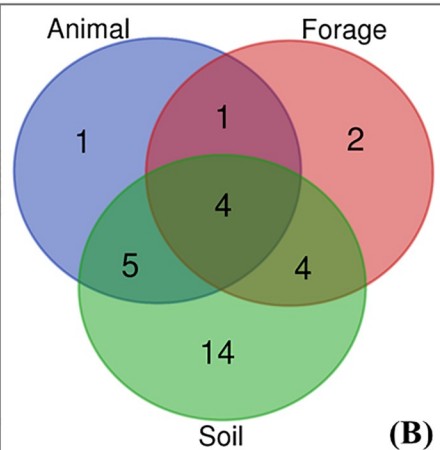 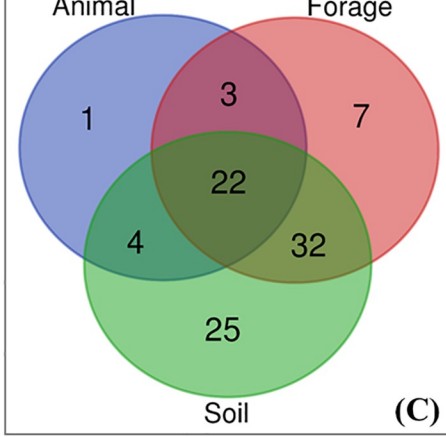

**Fig 7. Number of farms that conducted mineral tests on animal, forage and soil regardless of incidence of hypomagnesaemic tetany. (A)** Dairy farms, those that did not do neither test = 5. **(B)** Sheep farms, those that did not do neither test = 15. **(C)** Mixed beef and sheep farms, those that did not do neither test = 31 (c).

**Table 4. Fisher's exact test of association among mineral testing, incidence of presumed hypomagnesaemic tetany (PHT) and farm enterprises.**

| Farm enterprise | n | Exact significance (2-sided) | | |
|---|---|---|---|---|
| | | Animal | Forage | Soil |
| Beef-sheep | 127 | 1.000 | 0.157 | 0.852 |
| Dairy | 38 | 0.232 | 0.714 | 0.486 |
| Sheep | 46 | 0.074 | 0.137 | 1.000 |
| Total | 211 | 0.742 | 0.889 | 0.561 |

Presumed hypomagnesaemic tetany (PHT) was reported by 39% of the 285 cattle and sheep farmers who participated in this survey. This is greater than found by Roderick et al. [11] who reported 19% of dairy farms (n = 23), and 23% of suckler beef farms (n = 89) had presumed grass tetany under an organic livestock farming system in the UK. Equivalent information is not captured in any UK farm censuses, and official reporting is known to be low due to the widespread use of on-farm, non-veterinary diagnosis [18]. Our data indicate this problem may be affecting a small proportion of animals across a high proportion of farms. Further investigation is needed to establish if this remains the case with a representative survey and whether sub-clinical occurrence is being noticed by livestock managers. The prevalence of PHT was highest, at 55%, in mixed beef and sheep farms (n = 127) compared with 14% in sheep only farms (n = 46) in this survey. Further research is required to understand whether the higher prevalence of PHT in mixed livestock farm enterprises is due to factors such as the location of respondents, or is particular to the management systems being employed. The prevalence of PHT in dairy farm enterprises in this study (32%, n = 38) was higher than the 1982 report by Whitaker and Kelly [19], where an overall average incidence of 1% of clinical hypomagnesaemia in dairy farms, and 7% and 15% (n = 206) subclinical hypomagnesaemia in milking and dry dairy cows, respectively in England and Wales [19].

Unsurprisingly, the vast majority (96%) of farmers with PHT problems reported using an intervention, however, is noticeable that these were mainly free access and/or direct to animal supplements e.g., mineral blocks, boluses and salt licks. Only 3% (n = 9) of the respondents reported use of Mg-lime to correct underlying soil problems that may result in PHT. Other prevention measures such as pasture Mg-fertilisation, were not identified by participants. We did not capture whether animals survived the reported PHT, or were identified through diagnostics prior to visible symptoms. It is also noteworthy that 10 respondents reported use of an intervention which did not contain Mg, these were predominantly direct to animal trace element supplements (boluses, injections and drenches). This may be indicative of a misunderstanding that "mineral supplements" will contain the required micronutrient (here Mg), without further checking of the supplement composition.

Use of mineral tests on soil, forage or livestock were reported by 77% of participants. Testing was unrelated to the reported incidence of PHT. Soil tests were conducted by most participants (66%), whilst direct livestock measurements (e.g. blood/urine biomarker) were conducted by 24%. Although biomarker measurement increases animal handling time and can cause stress for the livestock, this may be a better way of detecting deficiencies of Mg than soil and forage tests, especially where urine data are used [13].

Hypomagnesaemic tetany in grazing ruminants is dependent not only on the quantity of Mg in the forage, but also the balance with herbage K, Ca, Na, ammonia, and rumen pH [2, 3, 20–22]. The forage tetany index (Eq 1), is often used as a forage mineral balance quality indicator [23], demonstrating the further potential value of forage analysis, which was only reported by 53% of respondents. Noteworthy in this study is that the majority of dairy businesses

reporting use of forage analysis had PHT. It is known that high K loading due to fertilisation of grazing pasturelands increases the risk of HypoMgT [24] and the current recommendation is to avoid using K fertilisers prior to turning livestock out in spring [25]. Opportunities to improve sward Mg-nutritional quality through breeding [26, 27] and pasture fertilisation or liming with Mg-lime may exist.

$$\textbf{\textit{Forage tetany index}} = \frac{\frac{K}{39.1}}{\left(\frac{Ca}{40.1} + \frac{Mg}{24.3}\right) * 2} \qquad (1)$$

*Where*, the elemental concentration (g kg$^{-1}$ dry weight) is divided by the atomic weight and multiplied by the valence of the respective element.

Index values are available for many soil properties, guiding pasture management decisions [25]. The majority of respondents in this study were from Wales and England (86%) where soil plant-available Mg concentrations are generally greater than the Index 2 (51 mg L$^{-1}$) which is used as a Mg-fertiliser recommendation threshold [28, 29]. It is noteworthy that grassland soils are often not optimally managed for pH, with 53% being below the recommended value of 6.0 in a recent private sector data synthesis [29]. This is consistent with our finding that approximately one in five participating farms were not using any diagnostic mineral tests, and 34% were not conducting soil mineral test, which could be compromising their ability to optimally manage pasture. Further work is needed to understand the relationship between the pasture soil Mg index value, variation in herbage Mg and the forage tetany index, and the occurrence of clinical and sub-clinical hypomagnesaemia. Interestingly, lower rates of Mg-deficiency, at 2%, (measured using a serum Mg) in Northern Ireland [10] may reflect the large swathe of the land area being underlain by Mg containing rocks, and thus soil naturally enriched with Mg [30].

Results from this study indicate that Mg deficiency is more widespread in UK grazing ruminant farms than reported previously. Mineral nutrition is influenced by a wide variety of factors, including composition of soils and forages, pasture management especially soil pH and nutrient management, the use of feeds/supplements, the type of livestock, as well as flock/herd wider health and physiological conditions. Further research to quantify the prevalence of clinical Mg deficiency, and to understand sustainable and effective mitigation measures, would be of benefit to grazing ruminant enterprises in the UK and elsewhere.

## Practical implications

The main implication of this work to the farmer is that they need to gather data on Mg in their pasture-livestock systems and undertake a dietary mineral audit [31]. The use of soil and forage mineral testing is more common than animal mineral testing, but none were universally used by the participants. It is clear that further information is needed on these inputs if hypomagnesaemic tetany (HypoMgT) risks are to be managed most effectively. In addition to Mg intake from forage, the intake from other sources also needs to be considered, which includes other feeds, supplements and water, and are covered in the full mineral audit process [31]. Care needs to be taken to ensure that supplements used are suitable to deliver the desired benefit. It may also be useful to assess the Mg status of the animal itself. Blood Mg is under tight homeostatic regulation and concentrations in serum/plasma will only decrease under extreme deficiency conditions. Urine Mg concentration is potentially a better indicator of an animal's Mg status and collection of urine can be less intrusive. During the mineral audit, it is also important to look at potential interacting elements, particularly K, which can reduce Mg absorption in the gut, and hence increase the incidence of HypoMgT. Soil Mg treatments are not widely used but could be part of strategies to reduce HypoMgT. However, balanced forage composition is important and farmers need to be aware of the unintended consequences of

using large quantities of K-containing fertilisers and farm yard manure for forage growth, which can then inhibit Mg absorption.

Whilst this paper has focused on Mg, many of these practical implications are expected to be relevant to a range of mineral micronutrients.

## Conclusion

More integrated soil-forage-animal assessments may improve the effectiveness of tackling hypomagnesaemic tetany, help highlight the root causes of the problem, and be used to guide optimal remediation/prevention strategies.

## Supporting information

**S1 Information. Semi-structured questionnaire and information sheet for interviewees.** (PDF)

**S1 Dataset. UK cattle and sheep farmers' responses with regards to hypomagnesaemic tetany.** "OriginalData" worksheet is the uncleaned version of the data. "CleanedData" worksheet is the cleaned data used for the analyses in this manuscript. (XLSX)

## Acknowledgments

The authors would like to thank XLVets, and the Agriculture and Horticulture Development Board (AHDB) who helped to distribute survey questionnaires, and all the farmers who participated in the survey.

## Author Contributions

**Conceptualization:** Diriba B. Kumssa, Beth Penrose, Peter A. Bone, J. Alan Lovatt, Martin R. Broadley, Nigel R. Kendall, E. Louise Ander.

**Data curation:** Diriba B. Kumssa, Beth Penrose, J. Alan Lovatt, Nigel R. Kendall.

**Formal analysis:** Diriba B. Kumssa, Nigel R. Kendall.

**Funding acquisition:** J. Alan Lovatt, Martin R. Broadley, Nigel R. Kendall, E. Louise Ander.

**Investigation:** J. Alan Lovatt, Martin R. Broadley, Nigel R. Kendall, E. Louise Ander.

**Methodology:** Diriba B. Kumssa, Beth Penrose, J. Alan Lovatt, Martin R. Broadley, Nigel R. Kendall, E. Louise Ander.

**Project administration:** Beth Penrose, Peter A. Bone, J. Alan Lovatt, Martin R. Broadley, Nigel R. Kendall, E. Louise Ander.

**Resources:** J. Alan Lovatt, Martin R. Broadley, Nigel R. Kendall, E. Louise Ander.

**Software:** Diriba B. Kumssa.

**Supervision:** Martin R. Broadley, Nigel R. Kendall, E. Louise Ander.

**Visualization:** Diriba B. Kumssa.

**Writing – original draft:** Diriba B. Kumssa, Beth Penrose, E. Louise Ander.

**Writing – review & editing:** Diriba B. Kumssa, Beth Penrose, Martin R. Broadley, Nigel R. Kendall, E. Louise Ander.

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
