## [Decision Letter · Decision Letter 0]

12 Jul 2019

PONE-D-19-17536

A reconnaissance survey of farmers’ awareness of hypomagnesaemic tetany in UK cattle and sheep farms

PLOS ONE

Dear Dr Kumssa,

Thank you for submitting your manuscript to PLOS ONE. After careful consideration, we feel that it has merit but does not fully meet PLOS ONE’s publication criteria as it currently stands. Therefore, we invite you to submit a revised version of the manuscript that addresses the points raised during the review process.

We would appreciate receiving your revised manuscript by Aug 26 2019 11:59PM. To enhance the reproducibility of your results, we recommend that if applicable you deposit your laboratory protocols in protocols.io, where a protocol can be assigned its own identifier (DOI) such that it can be cited independently in the future. For instructions see: http://journals.plos.org/plosone/s/submission-guidelines#loc-laboratory-protocols

We look forward to receiving your revised manuscript.

Kind regards,

Simon Russell Clegg, PhD

Academic Editor

PLOS ONE

Journal Requirements:

This work was supported by the Biotechnology and Biological Sciences Research Council (grant numbers BB/N004280/1 [ELA], BB/N004302/1 [MRB], BB/N004272/1 [AL]) and the Natural Environment Research Council, through the jointly funded initiative Sustainable Agriculture Research and Innovation Club (SARIC). The project is entitled Magnesium Network (MAG‐NET): Integrating Soil-Crop‐Animal Pathways to Improve Ruminant Health. https://gtr.ukri.org/projects?ref=BB%2FN004302%2F1

We note that one or more of the authors are employed by a commercial company: Livestock and Grassland Mineral Consultancy

Additional Editor Comments:

Many thanks for submitting your manuscript to PLOS One

Your manuscript was reviewed by two reviewers (finding expert reviewers in this area is difficult, but we managed to find two).

One recommended to reject the manuscript and the other recommended a major revision.

As this is a clear gap in knowledge and current research, I have recommended a major revision

It would be most helpful if you could respond to each individual comment, and modify the manuscript appropriately.

Wishing you the best of luck with the manuscript modifications

Thanks

Simon

Reviewers' comments:

Reviewer's Responses to Questions

**Comments to the Author**

1. Is the manuscript technically sound, and do the data support the conclusions?

Reviewer #1: Partly

Reviewer #2: Partly

2. Has the statistical analysis been performed appropriately and rigorously? 

Reviewer #1: No

Reviewer #2: I Don't Know

3. Have the authors made all data underlying the findings in their manuscript fully available?

Reviewer #1: Yes

Reviewer #2: Yes

4. Is the manuscript presented in an intelligible fashion and written in standard English?

Reviewer #1: Yes

Reviewer #2: Yes

5. Review Comments to the Author

Reviewer #1: This is an interesting topic, but the survey presented is very high level and simplistic. The questions were not tested or validated prior to use. The sample is not described as being representative of any major industry and geographic region.

There is no discussion of how many surveys were distributed or the response rate.

The statistical analysis of the main study questions does not account for the potential for confounding by enterprise type.

Where associations are examined, only p values are reported. There are no measures of association or 95%CI.

In the results, the authors jump between reporting percentages with and without a decimal place. Please be consistent. There also needs to be a clear numerator and denominator in each case or reference to a table with that information.

The number of figures needs to be reduced dramatically as most do not add any information to the manuscript and could be captured adequately in the text or tables.

The discussion starts with some of the study limitations and does not make a convincing case for what this paper adds to the literature.

Sward composition is mentioned in the abstract but not in the main results of the paper.

Reviewer #2: The results of the current study about hypomagnesemia have been obtained by a questionnaire with the usual limitations of such an approach. There are some recommendations.

1. Please integrate previous data about this topic from UK: Whitaker and Kelly (1982) Vet. Rec. 110, 450-451 and Wards and Parker (1999) Br. Soc. Anim. Sci. 24, 21-26.

2. The data from farms without hypomagnesemia should be omitted. These results are without primary interest. It could be mentioned very briefly in the discussion. Please focus on the results with hypomagnesemia. It appears to me that hypomagnesemia in dairy cattle is not the major problem at the present time.

3. Please restrict the discussion on epidemiology. Possible causes of hypomagnesemia should be reduced, because no available results of the current study can be presented.

4. I am missing a conclusion and a perspective for future research. Does it make sense to initiate corresponding measurements to confirm the data?

5. Why so many authors for a questionnaire?

6. Page 3 – line 52: Reference 5: Related to hypocalcaemia? Correct?

7. Page 4 – line 78: Underwood (number?)

8. Page 8 – table 2: * is not clear for me. What does it mean?

9. Page 11 – Equation 1: The equation should be explained with corresponding values. Where are the risks?

10. Page 11 – line 289: Kuusela et al.: not in the list of references

11. I am missing a paragraph about a relation of the own data with data from the literature. What is really knew?

6. PLOS authors have the option to publish the peer review history of their article (what does this mean?). If published, this will include your full peer review and any attached files.

Reviewer #1: No

Reviewer #2: No

---

## [Decision Letter · Decision Letter 1]

19 Sep 2019

PONE-D-19-17536R1

A reconnaissance survey of farmers’ awareness of hypomagnesaemic tetany in UK cattle and sheep farms

PLOS ONE

Dear Dr Kumssa,

Thank you for submitting your manuscript to PLOS ONE. After careful consideration, we feel that it has merit but does not fully meet PLOS ONE’s publication criteria as it currently stands. Therefore, we invite you to submit a revised version of the manuscript that addresses the points raised during the review process.

Many thanks for resubmitting your manuscript to PLOS One

It has been reviewed by one of same reviewers as previously, and a new reviewer who made some very minor suggestions, (mainly typos and grammatical errors). Both reviewers also suggest the addition of a small conclusion, and some advice for farmers.

If you could make the minor changes, and add in a small advice section and conclusion that will be great.

I would be grateful if you could make these small changes and resubmit it. Please do not worry about a rebuttal to the minor typo/ grammatical reviewers comments, they are so minor that just a line saying all comments were actions will be more than enough.

I will then recommend it for acceptance and publication without a need for re-review

Many thanks

Simon

We would appreciate receiving your revised manuscript by Nov 03 2019 11:59PM. To enhance the reproducibility of your results, we recommend that if applicable you deposit your laboratory protocols in protocols.io, where a protocol can be assigned its own identifier (DOI) such that it can be cited independently in the future. For instructions see: http://journals.plos.org/plosone/s/submission-guidelines#loc-laboratory-protocols

A very brief rebuttal letter that responds to the conclusion and advice section of the reviewer comments. The typos etc can just be a single line which says 'these were all actioned'. This letter should be uploaded as separate file and labeled 'Response to Reviewers'.A marked-up copy of your manuscript that highlights changes made to the original version. This file should be uploaded as separate file and labeled 'Revised Manuscript with Track Changes'.An unmarked version of your revised paper without tracked changes. This file should be uploaded as separate file and labeled 'Manuscript'.

We look forward to receiving your revised manuscript.

Kind regards,

Simon Russell Clegg, PhD

Academic Editor

PLOS ONE

Reviewers' comments:

Reviewer's Responses to Questions

**Comments to the Author**

1. If the authors have adequately addressed your comments raised in a previous round of review and you feel that this manuscript is now acceptable for publication, you may indicate that here to bypass the “Comments to the Author” section, enter your conflict of interest statement in the “Confidential to Editor” section, and submit your "Accept" recommendation.

Reviewer #2: All comments have been addressed

Reviewer #3: All comments have been addressed

2. Is the manuscript technically sound, and do the data support the conclusions?

Reviewer #2: Yes

Reviewer #3: Yes

3. Has the statistical analysis been performed appropriately and rigorously? 

Reviewer #2: I Don't Know

Reviewer #3: Yes

4. Have the authors made all data underlying the findings in their manuscript fully available?

Reviewer #2: Yes

Reviewer #3: Yes

5. Is the manuscript presented in an intelligible fashion and written in standard English?

Reviewer #2: Yes

Reviewer #3: Yes

6. Review Comments to the Author

Reviewer #2: I am still missing conclusions about future research according the present results. What should or can be done? Are there any advices for the farmers?

Reviewer #3: This is a very interesting manuscript regarding an issue which we do see on farms around the UK and further afield, but rarely. It is very nice to see some new research and literature in the area. As I have only been asked to review a revised version, I have a few comments but all are very minor (typos, grammatical etc), as I felt the manuscript was very good and well written. I commend you on that and wish you the best of luck with future research. I will keep an eye out for more work from your group.

Line 32- There was a highly significant

Line 55- although you define MypoMgT in the abstract, it would also be useful to define it early in the introduction as well, as some people (like me) don’t read the abstract until the end.

Line 61- same is true for defining PHT

Line 65- this is a little unclear, I do wonder if it is meant to read as hypomagnesaemia related conditions were seen in 1-4% of cows ….

Line 70-71- again unclear. I think it should read where magnesium, usually as part of a mineral premix, ….

Line 82- would the reference not be better written as Roderick et al., ?

Line 103- Should read- Prior to completing the survey, farmers were presented with an information sheet……

Line 108- Should read - English and Welsh, in both paper and digital form, and participants….

There is a stray ‘o’ in line 170- not sure what that should be

Line 174 – maybe should read as …’76% had used a similar range…’

Line 265 would the reference not be better written as Roderick et al., ?

Line 274. Sentence break needed …. In this survey. Further research ….

Line 278, comma after the reference may help flow

Line 288, a comma after ‘did not contain Mg,’ may aid flow

Line 307- systems may read better

I would also like to see a small paragraph as a bit of advice to farmers. As this is a relatively unresearched area, I think a bit of advice to farmers would be most beneficial

But overall it is a very well written manuscript, which I do not need to review again if all comments are addressed.

With very best wishes for the future

7. PLOS authors have the option to publish the peer review history of their article (what does this mean?). If published, this will include your full peer review and any attached files.

Reviewer #2: No

Reviewer #3: No

---

## [Author Response · Author response to Decision Letter 1]

25 Sep 2019

Response to reviewers document has been uploaded.

---

## [Editor Report · Decision Letter 2]

1 Oct 2019

A reconnaissance survey of farmers’ awareness of hypomagnesaemic tetany in UK cattle and sheep farms

PONE-D-19-17536R2

Dear Dr. Kumssa

We are pleased to inform you that your manuscript has been judged scientifically suitable for publication and will be formally accepted for publication once it complies with all outstanding technical requirements.

With kind regards,

Simon Russell Clegg, PhD

Academic Editor

PLOS ONE

Additional Editor Comments (optional):

Many thanks for resubmitting your manuscript to PLOS One and making the changes which were suggested

I am very grateful for your efforts, and I have recommended your manuscript for publication

I want to take this opportunity to thank you for your efforts and wish you the best of luck for your future research

Many thanks

Simon
---

## [Editor Report · Acceptance letter]

4 Oct 2019

PONE-D-19-17536R2 

A reconnaissance survey of farmers’ awareness of hypomagnesaemic tetany in UK cattle and sheep farms 

Dear Dr. Kumssa:

I am pleased to inform you that your manuscript has been deemed suitable for publication in PLOS ONE. Congratulations! Your manuscript is now with our production department. 

With kind regards,

on behalf of

Dr. Simon Russell Clegg 

Academic Editor

PLOS ONE